# A Progressively Expanded Database for Automated Lung Sound Analysis: An Update

**Fu-Shun Hsu** [1,2,3]**, Shang-Ran Huang** [3]**, Chien-Wen Huang** [4]**, Yuan-Ren Cheng** [3,5,6]**, Chun-Chieh Chen** [4]**, Jack Hsiao** [7] **, Chung-Wei Chen** [8] **and Feipei Lai** [1,]*****

[1] Graduate Institute of Biomedical Electronics and Bioinformatics, National Taiwan University, Taipei 10617, Taiwan; fshsu@heroic-faith.com
[2] Cardiovascular Intensive Care Unit, Far Eastern Memorial Hospital, New Taipei 22060, Taiwan
[3] Heroic Faith Medical Science Co., Ltd., Taipei 11049, Taiwan; shane.huang@heroic-faith.com (S.-R.H.); infie.cheng@heroic-faith.com (Y.-R.C.)
[4] Avalanche Computing Inc., Taipei 10687, Taiwan; cwhuang@avalanc.com (C.-W.H.); jaychen@avalanc.com (C.-C.C.)
[5] Department of Life Science, College of Life Science, National Taiwan University, Taipei 10617, Taiwan
[6] Institute of Biomedical Sciences, Academia Sinica, Taipei 11529, Taiwan
[7] HCC Healthcare Group, New Taipei 22060, Taiwan; jack.hsiao@hsiaohospital.org
[8] Division of Thoracic Surgery, Department of Surgery, Far Eastern Memorial Hospital, New Taipei 22060, Taiwan; femhcs@gmail.com
***** Correspondence: flai@ntu.edu.tw

**Featured Application: Auscultatory lung sound analysis in healthcare.**

**Abstract:** We previously established an open-access lung sound database, HF_Lung_V1, and developed deep learning models for inhalation, exhalation, continuous adventitious sound (CAS), and discontinuous adventitious sound (DAS) detection. The amount of data used for training contributes to model accuracy. In this study, we collected larger quantities of data to further improve model performance and explored issues of noisy labels and overlapping sounds. HF_Lung_V1 was expanded to HF_Lung_V2 with a 1.43× increase in the number of audio files. Convolutional neural network–bidirectional gated recurrent unit network models were trained separately using the HF_Lung_V1 (V1_Train) and HF_Lung_V2 (V2_Train) training sets. These were tested using the HF_Lung_V1 (V1_Test) and HF_Lung_V2 (V2_Test) test sets, respectively. Segment and event detection performance was evaluated. Label quality was assessed. Overlap ratios were computed between inhalation, exhalation, CAS, and DAS labels. The model trained using V2_Train exhibited improved performance in inhalation, exhalation, CAS, and DAS detection on both V1_Test and V2_Test. Poor CAS detection was attributed to the quality of CAS labels. DAS detection was strongly influenced by the overlapping of DAS with inhalation and exhalation. In conclusion, collecting greater quantities of lung sound data is vital for developing more accurate lung sound analysis models.

**Keywords:** auscultation; convolutional neural network; deep learning; gated recurrent unit; lung sound



## 1. Introduction

Respiration is an essential vital sign. Changes in the frequency or intensity of respiratory lung sounds and the manifestation of continuous adventitious sounds (CASs), such as wheezes, stridor, squawk, gasp, and rhonchi, and discontinuous adventitious sounds (DASs), such as crackles and pleural friction rubs, are associated with pulmonary disorders, including pneumonia, asthma, chronic obstructive pulmonary disease (COPD), and cystic fibrosis [1,2]. Detection of adventitious sounds during lung sound auscultation helps health-care professionals make clinical decisions. However, the interpretation of lung sounds auscultated with a traditional stethoscope involves inherent subjectivity and

relies on auscultation capability [3]. Electronic auscultation and computerized lung sound analysis can facilitate the interpretation and overcome the above limitations [4].

Computerized lung sound analysis methods based on machine learning have been comprehensively reviewed [5–7]. Some previous machine learning studies focused on classifying individuals into a healthy group or groups with specific types of respiratory disorders according to their lung sounds. Some studies developed algorithms to classify normal lung sounds and various types of adventitious lung sounds. Some studies proposed methods to detect breath phases and adventitious sounds in lung sound recordings. Over the last few years, more and more researchers have turned their attention to using deep learning for computerized lung sound analysis. Various architectures of deep neural networks (DNN), such as convolutional neural network (CNN) [8–18], recurrent neural network [9,13,18–21], residual neural network [22,23], temporal convolutional network [24], regional CNN [25], and DNN with attention mechanism [26,27] have been investigated in developing automated lung sound analysis systems. However, many studies were limited by small data volumes, and some datasets used in these studies were private so a comparison between the proposed models cannot be realized. As data size plays an important role to train a more accurate deep learning model, a larger open-access dataset can greatly help the development of lung sound analysis methods.

Some public databases of respiratory sound have been established. Rocha et al. [28] established an open-access database with which new algorithms for respiratory sound classification can be evaluated and compared. This database, compiled for the first scientific challenge of the International Federation for Medical and Biological Engineering International Conference on Biomedical and Health Informatics (ICBHI) 2017, included 920 audio recordings from 126 subjects and text files showing disease diagnosis and precise location of respiratory cycles, wheezes, and crackles. In addition, Fraiwan and colleagues [29] established a dataset of lung sounds recorded from healthy and unhealthy subjects which included disease labels, such as asthma, heart failure, pneumonia, bronchitis, and COPD. They focused on designing automated machine learning algorithms for the detection of pulmonary disorders. Previously, we reported the development of the world's largest open-access lung sound database, HF_Lung_V1 (https://gitlab.com/techsupportHF/HF_Lung_V1, accessed on 18 January 2022). Different from the ICBHI 2017 database, we not only labeled respiratory cycles, wheezes, and crackles, but also labeled inhalations, exhalations, stridor, and rhonchi. However, the HF_Lung_V1 database did not include labels of disease diagnosis.

HF_Lung_V1 is benchmarked with several variants of recurrent neural networks [18]. However, while the HF_Lung_V1 database shows high potential for exploiting deep learning for automated inhalation, exhalation, CAS, and DAS detection, it remains to be further improved to achieve better accuracy. Because the performance of deep learning models is positively associated with the size of the training set [30], an increase in the number of sound recordings and labels is expected to improve the database. In addition, given the possible presence of noisy labels, label quality needs to be further optimized. Therefore, the aim of this study was to present an updated database, HF_Lung_V2, which is based on HF_Lung_V1 but expanded to include more high-quality lung sound files and labels. The performance of the detection model trained with HF_Lung_V2 was also evaluated.

## 2. Materials and Methods

### 2.1. Participants

We acquired new lung sound recordings from two sources to form a new dataset, HF_Lung_V1_incremental_package (HF_Lung_V1_IP), which can be downloaded at https://gitlab.com/techsupportHF/HF_Lung_V1_IP (accessed on 13 April 2022). By adding the lung sound recordings in HF_Lung_V1_IP with those in HF_Lung_V1, we could establish the lung sound database, HF_Lung_V2.

In HF_Lung_V1_IP, the lung sounds from the first source were collected from seven inpatients receiving long-term mechanical ventilation support in a respiratory care ward (RCW) or respiratory care center (RCC) between October 2019 and December 2019. The lung

sounds from the second source were obtained from another 32 inpatients with adventitious sounds at Far Eastern Memorial Hospital (FEMH), Taipei, Taiwan between January 2019 and November 2019. All participants were Taiwanese and aged ≥20 years.

Table 1 summarizes the demographic data and the characteristics of respiratory disease of the subjects in HF_Lung_V2. Note that not every FEMH inpatient was with respiratory disease. We did not have the information about the subjects whose lung sound recordings were acquired and provided by the Taiwan Society of Emergency and Critical Care Medicine (TSECCM). The information about subjects' other co-occurring diseases can be found in Table S1.

**Table 1.** Demographic data and the characteristics of respiratory diseases of the subjects in HF_Lung_V2.

| | Subjects from RCC/RCW | Subjects from FEMH | Subjects in TSECCM Database |
|---|---|---|---|
| Number (n) | 25 | 32 | 243 |
| Sex (M/F/NA) | 16/8/1 | 19/13 | NA |
| Age | 64.5 ± 18.5 | 68.2 ± 11.7 | NA |
| Height (cm) | 162.3 ± 9.1 * | 160.0 ± 8.8 | NA |
| Weight (kg) | 63.4 ± 12.7 * | 59.7 ± 12.3 | NA |
| BMI (kg/m$^2$) | 24.0 ± 4.3 * | 23.6 ± 4.6 | NA |
| Recording device | HF-Type-1 & Littmann 3200 | Littmann 3200 | Littmann 3200 |
| Respiratory disease | | | |
| Acute exacerbation of chronic obstructive pulmonary disease | 1 (4%) | 1 (3%) | NA |
| Acute respiratory distress syndrome | 1 (4%) | 0 (0%) | NA |
| Acute respiratory failure | 4 (16%) | 0 (0%) | NA |
| Asthma | 0 (0%) | 1 (3%) | NA |
| Bronchitis | 1 (4%) | 0 (0%) | NA |
| Chronic respiratory failure | 12 (48%) | 2 (6%) | NA |
| Chronic obstructive pulmonary disease | 2 (8%) | 5 (16%) | NA |
| Emphysema | 1 (4%) | 0 (0%) | NA |
| Pleural effusion | 0 (0%) | 1 (3%) | NA |
| Pneumoconiosis | 0 (0%) | 1 (3%) | NA |
| Pneumonia | 6 (24%) | 7 (22%) | NA |
| Pulmonary embolism | 0 (0%) | 1 (3%) | NA |

* With four missing data. NA: not available.

### 2.2. Lung Sound Recordings

We used two different devices to record the participants' lung sounds. The first device was a commercial handheld electronic stethoscope, Littmann 3200 (3M, Saint Paul, MN, USA); the second one was a customized auscultation device, HF-Type-1 [18], which supported multichannel continuous sound recording through attachable acoustic sensors. We followed the same recording protocols described in our previous study to acquire the lung sounds from the participants [18]. Eight auscultation locations (Figure 1a) were firstly defined—L1: second intercostal space (ICS) on the right midclavicular line (MCL); L2: fifth ICS on the right MCL; L3: fourth ICS on the right midaxillary line (MAL); L4: tenth ICS on the right MAL; L5: second ICS on the left MCL; L6: fifth ICS on the left MCL; L7: fourth ICS on the left MAL; and L8: tenth ICS on the left MAL. Briefly, when the Littmann

3200 device was used, to complete a standard round of recording, lung sounds must be auscultated sequentially from L1 to L8 (white arrows in the top graph in Figure 1b). When the HF-Type-1 device was used, acoustic sensors were attached at L1, L2, L4, L5, L6, and L8, and lung sounds were continuously recorded at the same time. A standard round of lung sound recording with the HF-Type-1 device consisted of the acquisition of continuous signals from each of these six locations over 30 min (white arrows in the bottom graph in Figure 1b).

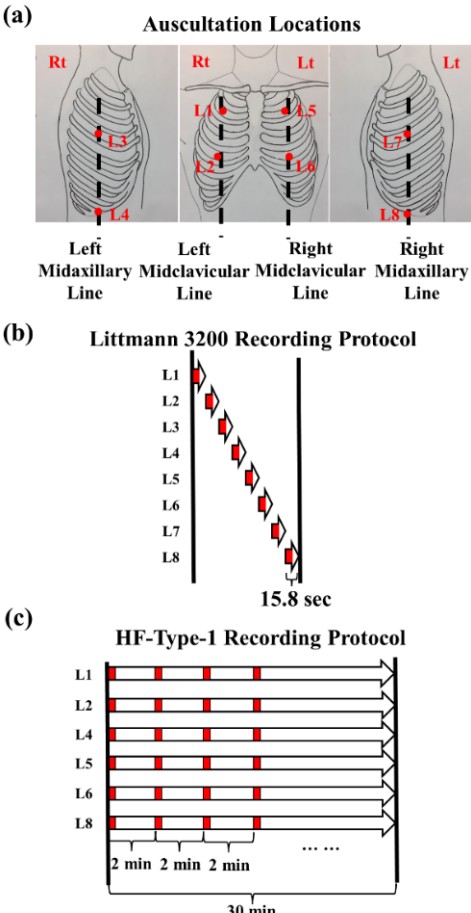

**Figure 1.** Lung sound auscultation locations and protocol. (**a**) Predefined auscultation locations (L1–L8). (**b**) Littmann 3200 recording protocol. (**c**) HF-Type-1 recording protocol. In (**b**,**c**), white right arrows and red blocks represent original continuous recordings and truncated 15-s recordings, respectively. This figure is adapted from our previous study under a Creative Commons Attribution (CC BY) license [18].

The lung sounds of the seven RCW/RCC inpatients were obtained with the Littmann 3200 device for 4–5 rounds and the HF-Type-1 device for 3–4 rounds. The lung sounds of the 32 inpatients at FEMH were recorded using the Littmann 3200 device alone for 1–3 rounds. The sampling rate of the recording devices was 4000 Hz, and the bit depth was 16 bits. We did not ask the participants to take deep breaths during sound recording.

### 2.3. Data Preparation and Labeling

The length of audio files recorded with the Littmann 3200 device was originally set at 15.8 s; as such, the terminal 0.8-s segment was deleted to obtain 15-s audio files (red blocks in Figure 1b). A standard round of continuous recordings of lung sound generated by the HF-Type-1 device was longer than 30 min; therefore, we truncated the first 15-s segment of every 2-min signal in the continuous recordings for subsequent analysis (red blocks in Figure 1b).

Two licensed respiratory therapists with 8 and 4 years of clinical experience, respectively, and one registered nurse with 13 years of clinical experience performed the labeling. The labelers' licenses were granted by the Ministry of Examination, Taiwan, and they gained their clinical experience in Taiwan. Each lung sound file was labeled by only one individual, but regular consensus meetings were held to evaluate the labeling and reach a consensus among all labelers. A self-developed software [31] was used to do the labeling. During labeling, the labelers first used the software to load a 15-s lung sound recording (raw signal displayed in Figure 2a), and the corresponding spectrogram (green square in Figure 2b) would be displayed subsequently. The labelers were asked to listen to the sound and observe its corresponding temporal-spectral patterns on the spectrogram once or twice first. Then, they must label all the inhalations and exhalations in the 15-s recording. The labeler can move the yellow lines (indicated by white arrows in Figure 2b) to define a region of interest (ROI) (transparent yellow patch in Figure 2b). By clicking the area of which start and end times were bounded by the ROI on specific label tracks (red arrows in Figure 2b), the labelers can generate corresponding labels. After having the labels of inhalations (I) and exhalations (E), they can proceed to label wheezes, stridor, rhonchi, and DASs. When labeling DASs, they were asked to label the start and end times of a series of crackle sounds but not label each explosive sound; additionally, the sounds of pleural friction rubs were not included in the DAS labels (D) in this study. During the labeling process, they can replay the sound and make a modification to the labels as many times as they wanted. They were instructed not to label if a sound was inaudible. If a sound was cut off at the very beginning or end of a recording which made them have less confidence in labeling, they can skip the labeling of it. After labeling, the software automatically created a label file. The output of the label started with the type of label (I, E, Wheeze, Stridor, Rhonchi, or D), and it was followed by the start and end times of the event, shown in Figure 2c.

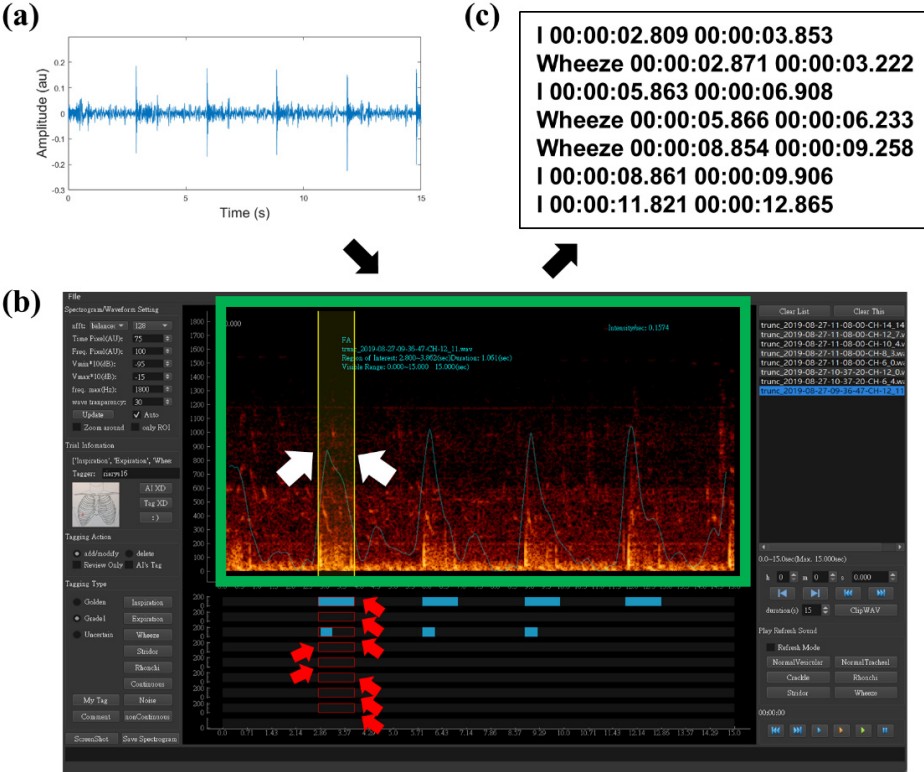

**Figure 2.** Labeling process. (**a**) Raw signal of a lung sound recording. (**b**) Labeling software. (**c**) Output in a label file. In (**b**), the green square represents the spectrogram, the yellow lines indicated by the white arrows represent the boundaries of an ROI (the transparent yellow patch), and the red arrows indicate areas on different label tracks of which start and end times bounded by the ROI; when a labeler clicks on one of them, a corresponding label is generated.

Note that during deep-learning processing, labels of wheezes (W), stridor (S), and rhonchi (R) were combined to obtain CAS labels (C).

### 2.4. Dataset Arrangement

Because we recorded several rounds of the lung sounds from one individual within a short period of time (few hours), the sound patterns collected at the same location may bear similarity. Therefore, to fairly evaluate the performance of the trained models, we put the lung sound recordings collected from the same subject to either training data set or test data set. The ratio of the numbers of recordings in the training and test set was maintained at approximately 4:1.

### 2.5. Benchmarking of HF_Lung_V2

The convolutional neural network with bidirectional gated recurrent unit (CNN–BiGRU) model is presented in Figure 3. The formulas of ReLU [32] and Sigmoid [32] layers are as follows:

$$\text{ReLU}(x) = \max(x, 0), \tag{1}$$

$$\text{Sigmoid}(x) = 1/(1 + \exp(-x)). \tag{2}$$

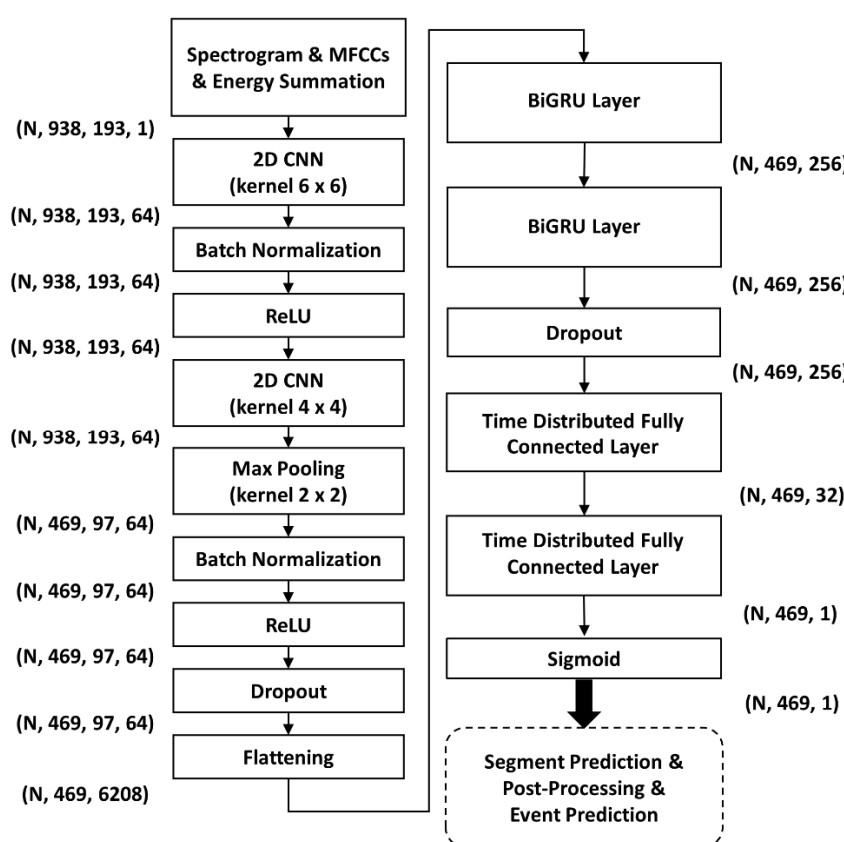

**Figure 3.** Architecture of the CNN–BiGRU model. This figure is adapted from our previous study under a Creative Commons Attribution (CC BY) license [18].

The CNN–BiGRU model outperformed all benchmark models used in our previous study [18]. Therefore, the model was used to benchmark HF_Lung_V2 in the present study.

Figure 4 shows the pipeline for preprocessing, deep-learning processing, and post-processing. All the codes were written using Python 3.7. During the preprocessing, we first used a 10th-order Butterworth high-pass filter with a cutoff frequency of 80 Hz to process the raw 15-s signal. Then, we computed the spectrogram of the filtered signal using short-time Fourier transform (nfft = 256, window length = 256, window type = hann, overlap = 0.75). The output of the short-time Fourier transform was a 938 × 129 array,

where 938 was the number of time segments and 129 was the number of frequency bins. Next, Mel-frequency cepstral coefficients (MFCCs) [33], including 20 static, 20 delta, and 20 delta–delta coefficients were calculated for each time segment. Finally, we calculated energy summation in four frequency bands, 0–250, 250–500, 500–1000, and 0–2000 Hz, for each time segment. Min–max normalization was applied to each of the spectrogram, MFCCs, and energy summation. Then, the concatenation of normalized spectrogram, MFCCs, and energy summation, a 938 × 193 array, were fed as input to train the CNN-BiGRU model and generate the results of segment prediction. We used Tensorflow 2.10 framework to construct the layers of neural networks. During training, we used Adam as the optimizer, and batch size was set to 64. We set the initial learning rate to 0.0001 with a step decay (0.2×) when the validation loss did not decrease for 10 epochs. The learning process stopped when no improvement occurred over 50 consecutive epochs. The training was done on an Ubuntu 18.04 server provided by the National Center for High-Performance Computing in Taiwan (Taiwan Computing Cloud (TWCC)). It was equipped with an Intel(R) Xeon(R) Gold 6154 @3.00 GHz CPU with 90 GB RAM. We used the CUDA 10, and CuDNN 7 library to execute the training and inference on an NVIDIA Titan V100 graphics card for GPU acceleration. Finally, we refined the results of segment prediction by merging the connected segments with positive prediction and those with an interval smaller than 0.5 s to generate the detected events. Subsequently, the events with a duration smaller than 0.05 s were seen as burst noises and removed. More details can be found in our previous study [18].

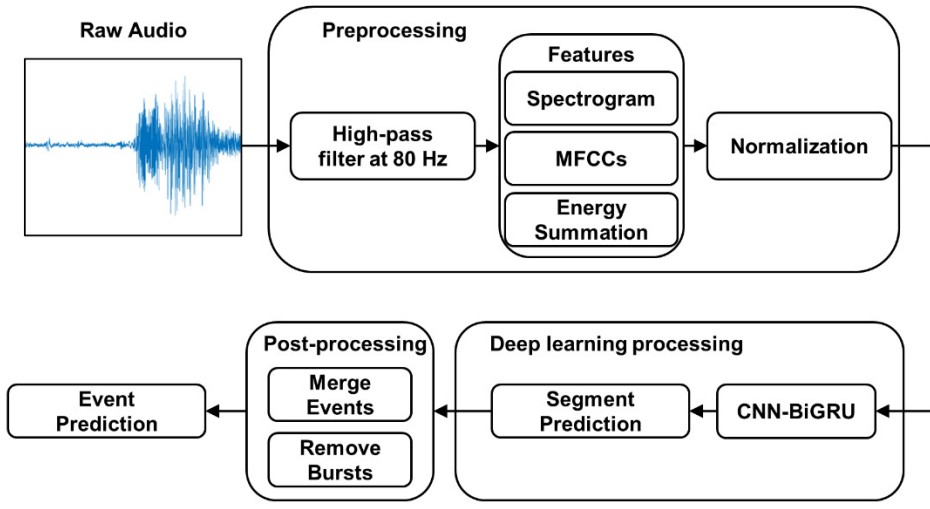

**Figure 4.** Pipeline for preprocessing, deep-learning processing, and postprocessing. Detailed information is described in our previous publication [18]. This figure is adapted from our previous study under a Creative Commons Attribution (CC BY) license [18].

Subject-wise five-fold cross-validation was conducted during the training process. To obtain more stable results, this procedure was repeated thrice. To mitigate data imbalance, we used the audio files containing at least one event of CASs and DASs to train and test the respective models.

*2.6. Performance Evaluation*

Figure 5 illustrates how we evaluated the performance of segment- and event-detection. First, the ground-truth event labels (red horizontal bars in Figure 5a) were used to generate the ground-truth time segments (red vertical bars in Figure 5b). Next, after we conducted binary thresholding on the output of the model to determine if the probability was high enough to be viewed as a detected segment (a blue vertical bar in Figure 5c), we could obtain the results of segment prediction. By comparing the results of segment prediction (blue vertical bars in Figure 5c) with the ground-truth time segments (Figure 5b), we defined true-positive (TP; orange vertical bars in Figure 5e), true-negative (TN; green vertical bars

in Figure 5e), false-positive (FP; black vertical bars in Figure 5e), and false-negative (FN; yellow vertical bars in Figure 5e) time segments.

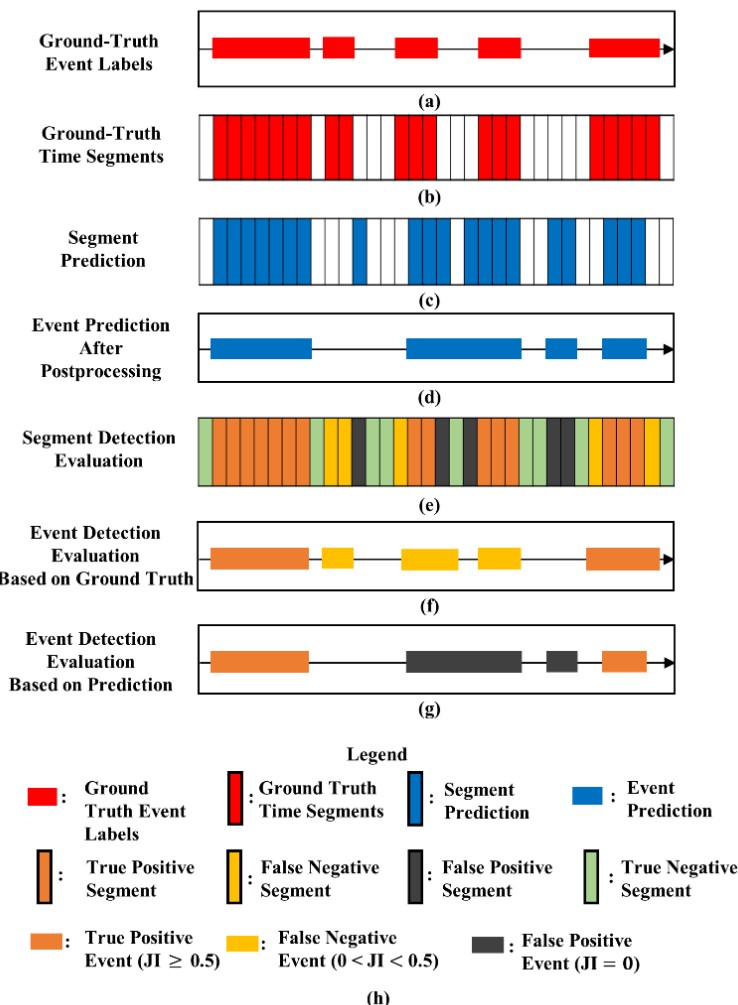

**Figure 5.** Illustration of segment- and event-detection evaluation: (**a**) ground-truth event labels, (**b**) ground-truth time segments, (**c**) segment prediction, (**d**) event prediction after postprocessing, (**e**) segment detection evaluation, (**f**) event evaluation based on ground-truth event labels, (**g**) event detection evaluation based on prediction, and (**h**) legend. JI: Jaccard index. This figure is adapted from our previous study under a Creative Commons Attribution (CC BY) license [18].

For event detection, the segment prediction results were post-processed (see Section 2.5) to obtain the event prediction results (Figure 5d). Ground-truth event labels and the results of event prediction were alternately employed as references to evaluate the accuracy of event detection (evaluation using ground-truth labels as reference is displayed in Figure 5f, and evaluation using the results of event prediction as reference is displayed in Figure 5g). The Jaccard index (JI) [34], defined as follows,

$$\text{Jaccard Index (JI)} = \frac{(\text{target event} \cap \text{referenced event})}{(\text{target event} \cup \text{referenced event})}, \tag{3}$$

was used to evaluate the extent to which a target event overlapped with a referenced event. We defined a JI of $\geq 0.5$ as a correct match for generating a TP event (orange horizontal bars in Figure 5f,g). If a ground-truth event did not have a correctly matched detected event, it was defined as an FN event (yellow horizontal bars in Figure 5f). If a detected event did not have a correctly matched ground-truth event, it was considered an FP event (black horizontal bars in Figure 5g). A TN event cannot be defined in the event detection

evaluation process. Because the TP events were double-counted (orange horizontal bars in Figure 5f,g), a pair of TP events was viewed as a single TP event. However, all FN (Figure 5f) and FP events (Figure 5g) events were taken into computing the model performance even though some FN events overlapped with some FP events, which may have led to slightly biased results.

Accuracy, sensitivity, specificity, positive predictive value (PPV), and F1 score were computed to evaluate the performance of segment detection. Additionally, a receiver operating characteristic (ROC) curve was plotted, and the corresponding 95% confidence interval was displayed to help visualize the performance of segment detection between models. Area under the ROC curve (AUC) was calculated. Event detection was assessed using sensitivity, PPV, and F1 score. We selected a threshold generating the best accuracy of segment detection to calculate all the other performance indexes. The definitions of the performance indexes are described as follows:

$$\text{accuracy} = (TP + TN)/(TP + TN + FP + FN), \tag{4}$$

$$\text{sensitivity} = TP/(TP + FN), \tag{5}$$

$$\text{specificity} = (TN)/(TN + FP), \tag{6}$$

$$\text{PPV} = TP/(TP + FP), \tag{7}$$

$$\text{F1 score} = 2 \times (\text{sensitivity} \times \text{PPV})/(\text{sensitivity} + \text{PPV}). \tag{8}$$

Performance measurements were averaged and reported for 15 models obtained from the repeated five-fold cross-validation. The measures of accuracy, sensitivity, specificity, PPV, and F1 score between the models trained using V1_Train and V2_Train were statistically compared using a Student's *t*-test, while the AUC was compared using a Mann–Whitney U test.

### 2.7. Investigation of Label Quality and Sound Overlapping

Two of the various factors indicated in our previous study may have influenced model performance and thereby required investigation [18]: noisy labels and sound overlapping. First, label quality was reviewed to evaluate what types of patterns may lead to noisy labels. Second, the density of the start and end times of each type of I, E, C, and D labels in the V2_Train and V2_Test was calculated and plotted. Subsequently, the overlap ratios between I, E, C, and D labels were calculated.

## 3. Results

### 3.1. Summary of HF_Lung_V1 and HF_Lung_V2 Databases

Statistical data of lung sound files and labels employed in HF_Lung_V1 and HF_Lung_V2 are presented in Table 2. The number of participants increased from 261 in HF_Lung_V1 to 300 in HF_Lung_V2. Moreover, HF_Lung_V2 had 1.43 times the number of 15-s records (13,957) compared to those in HF_Lung_V1 (9765), and the total duration increased from 2441.25 min to 3489.25 min. In addition, the number of I, E, C, and D labels increased from 34,095 to 49,373, from 18,349 to 24,552, from 13,883 to 21,558, and from 15,606 to 19,617, respectively.

The statistics of the lung sound files and labels grouped by data sets can be found in Table 3. The numbers of 15-s files in the training and test data sets in HF_Lung_V1 were 7809 and 1956, respectively; and those in the training and test data sets in HF_Lung_V2 increased to 10554 and 3403, respectively. The number of 15-s files recorded with the Littmann 3200 device increased from 4504 in HF_Lung_V1 to 5156 in HF_Lung_V2, and those recorded with the HF-Type-1 device increased from 5261 in HF_Lung_V1 to 8801 in HF_Lung_V2 (Table S2).

**Table 2.** Statistical data of lung sound files in HF_Lung_V1 and HF_Lung_V2 databases.

| Database | | HF_Lung_V1 | HF_Lung_V2 |
|---|---|---|---|
| Subjects | | 261 | 300 |
| No. of 15-s recordings | | 9765 | 13,957 |
| Total duration (min) | | 2441.25 | 3489.25 |
| Inhalation | No.<br>Duration (min)<br>Mean (s) | 34,095<br>528.14<br>0.93 | 49,373<br>785.48<br>0.95 |
| Exhalation | No.<br>Duration (min)<br>Mean (s) | 18,349<br>292.85<br>0.96 | 24,552<br>374.24<br>0.91 |
| CAS | No. C/W/S/R<br>Duration (min) C/W/S/R<br>Mean (s) C/W/S/R | 13,883/8457/686/4740<br>191.16/119.73/9.46/61.98<br>0.83/0.85/0.83/0.78 | 21,558/13,139/914/7505<br>292.85/186.97/12.82/93.06<br>0.82/0.85/0.84/0.74 |
| DAS | No.<br>Duration (min)<br>Mean (s) | 15,606<br>230.87<br>0.89 | 19,617<br>281.55<br>0.86 |

CAS/C: continuous adventitious sound, DAS: discontinuous adventitious sound. W: wheeze, S: stridor, R: rhonchus.

**Table 3.** Statistics of lung sound files and labels in the HF_Lung_V1 and HF_Lung_V2 databases grouped by data sets.

| Database | | HF_Lung_V1 | | HF_Lung_V2 | |
|---|---|---|---|---|---|
| **Data Set** | | **Training** | **Test** | **Training** | **Test** |
| No. of 15-s recordings | | 7809 | 1956 | 10,554 | 3403 |
| Total duration (min) | | 1952.25 | 489 | 2638.5 | 850.75 |
| I | No. | 27,223 | 6872 | 39,057 | 10,316 |
| | Duration (min) | 422.17 | 105.97 | 623.02 | 162.46 |
| | Mean (s) | 0.93 | 0.93 | 0.96 | 0.94 |
| E | No. | 15,601 | 2748 | 18,334 | 6218 |
| | Duration (min) | 248.05 | 44.81 | 292.88 | 81.37 |
| | Mean (s) | 0.95 | 0.98 | 0.96 | 0.79 |
| C | No. C/W/S/R | 11,464/7027/657/3780 | 2419/1430/29/960 | 17,361/11,453/866/5042 | 4197/1686/48/2463 |
| | Duration (min) C/W/S/R | 160.16/100.71/9.10/50.35 | 31.01/19.02/0.36/11.63 | 240.40/163.77/12.34/64.29 | 52.45/23.19/0.48/28.77 |
| | Mean (s) C/W/S/R | 0.84/0.86/0.83/0.80 | 0.77/0.80/0.74/0.73 | 0.83/0.86/0.85/0.77 | 0.75/0.83/0.60/0.70 |
| D | No. | 13,794 | 1812 | 14,239 | 5378 |
| | Duration (min) | 203.59 | 27.29 | 210.96 | 70.59 |
| | Mean (s) | 0.89 | 0.90 | 0.89 | 0.79 |

I: inhalation labels, E: exhalation labels, C: continuous adventitious sound labels, D: discontinuous adventitious sound labels, W: wheeze labels, S: stridor labels, and R: rhonchus labels.

### 3.2. Performance Benchmark

Performance measurement of the CNN-BiGRU models in segment and event detection on V1_Test and V2_Test is presented in Table 4. We can see that the models trained using V2_Train performed significantly better than those trained using V1_Train in terms of most measures in the segment- and event-detection of inhalation and CAS on V1_Test or V2_Test.

The improvement was also significant in the segment- and event-detection of exhalation on V1_Test. However, the models trained using V2_Train only showed slightly improved performance in event detection of DAS on V1_Test. Superior performance of the models trained using V2_Train in inhalation, exhalation, and CAS detection were shown in the ROC curves derived from V1_Test (Figure 6a–c) and V2_Test (Figure 6e–g). However, the ROC curves of the models greatly overlapped with each other in DAS detection (Figure 6d,h).

**Table 4.** Performance measurement of CNN-BiGRU models trained using V1_Train and V2_Train and tested on V1_Test and V2_Test.

| | Segment Detection | | | | | | Event Detection | | |
|---|---|---|---|---|---|---|---|---|---|
| | Accuracy | PPV | Sensitivity | Specificity | F1 Score | AUC | PPV | Sensitivity | F1 Score |
| Inhalation | | | | | | | | | |
| Train_V1 on Test_V1 | 0.915 ± 0.001 | 0.833 ± 0.006 | 0.781 ± 0.010 | 0.954 ± 0.003 | 0.806 ± 0.004 | 0.962 ± 0.001 | 0.818 ± 0.005 | 0.855 ± 0.009 | 0.836 ± 0.005 |
| Train_V2 on Test_V1 | 0.921 ± 0.001 *** | 0.849 ± 0.005 *** | 0.794 ± 0.010 *** | 0.959 ± 0.002 *** | 0.821 ± 0.004 *** | 0.968 ± 0.001 *** | 0.836 ± 0.005 *** | 0.864 ± 0.009 ** | 0.850 ± 0.005 *** |
| Train_V1 on Test_V2 | 0.927 ± 0.002 | 0.834 ± 0.009 | 0.789 ± 0.010 | 0.961 ± 0.003 | 0.811 ± 0.005 | 0.970 ± 0.001 | 0.819 ± 0.007 | 0.864 ± 0.009 | 0.842 ± 0.005 |
| Train_V2 on Test_V2 | 0.931 ± 0.003 ††† | 0.844 ± 0.011 † | 0.799 ± 0.012 † | 0.963 ± 0.003 | 0.821 ± 0.008 ††† | 0.973 ± 0.002 ††† | 0.832 ± 0.010 ††† | 0.869 ± 0.011 | 0.851 ± 0.009 †† |
| Exhalation | | | | | | | | | |
| Train_V1 on Test_V1 | 0.875 ± 0.003 | 0.750 ± 0.013 | 0.536 ± 0.020 | 0.957 ± 0.004 | 0.625 ± 0.013 | 0.901 ± 0.003 | 0.604 ± 0.017 | 0.629 ± 0.022 | 0.616 ± 0.018 |
| Train_V2 on Test_V1 | 0.882 ± 0.004 *** | 0.763 ± 0.011 ** | 0.573 ± 0.026 *** | 0.957 ± 0.003 | 0.654 ± 0.018 *** | 0.912 ± 0.005 *** | 0.626 ± 0.026 * | 0.652 ± 0.025 * | 0.639 ± 0.025 ** |
| Train_V1 on Test_V2 | 0.923 ± 0.002 | 0.782 ± 0.009 | 0.630 ± 0.018 | 0.971 ± 0.002 | 0.698 ± 0.011 | 0.951 ± 0.002 | 0.751 ± 0.011 | 0.752 ± 0.020 | 0.752 ± 0.015 |
| Train_V2 on Test_V2 | 0.924 ± 0.003 | 0.783 ± 0.011 | 0.644 ± 0.019 | 0.970 ± 0.002 | 0.706 ± 0.012 | 0.954 ± 0.002 †† | 0.754 ± 0.017 | 0.749 ± 0.019 | 0.751 ± 0.017 |
| CAS | | | | | | | | | |
| Train_V1 on Test_V1 | 0.852 ± 0.004 | 0.683 ± 0.014 | 0.448 ± 0.031 | 0.950 ± 0.005 | 0.540 ± 0.022 | 0.870 ± 0.008 | 0.470 ± 0.025 | 0.423 ± 0.029 | 0.446 ± 0.026 |
| Train_V2 on Test_V1 | 0.869 ± 0.003 *** | 0.702 ± 0.007 *** | 0.566 ± 0.030 *** | 0.942 ± 0.004 | 0.626 ± 0.018 *** | 0.911 ± 0.005 *** | 0.539 ± 0.025 *** | 0.489 ± 0.030 *** | 0.514 ± 0.027 *** |
| Train_V1 on Test_V2 | 0.873 ± 0.002 | 0.659 ± 0.010 | 0.383 ± 0.032 | 0.963 ± 0.004 | 0.484 ± 0.024 | 0.890 ± 0.006 | 0.379 ± 0.028 | 0.339 ± 0.030 | 0.359 ± 0.028 |
| Train_V2 on Test_V2 | 0.884 ± 0.003 ††† | 0.675 ± 0.013 ††† | 0.495 ± 0.028 ††† | 0.956 ± 0.004 | 0.571 ± 0.018 ††† | 0.918 ± 0.005 ††† | 0.438 ± 0.023 ††† | 0.403 ± 0.028 ††† | 0.420 ± 0.025 ††† |
| DAS | | | | | | | | | |
| Train_V1 on Test_V1 | 0.828 ± 0.005 | 0.733 ± 0.016 | 0.694 ± 0.022 | 0.888 ± 0.012 | 0.713 ± 0.009 | 0.894 ± 0.004 | 0.624 ± 0.017 | 0.593 ± 0.016 | 0.608 ± 0.014 |
| Train_V2 on Test_V1 | 0.829 ± 0.006 | 0.728 ± 0.016 | 0.706 ± 0.017 | 0.883 ± 0.011 | 0.717 ± 0.010 | 0.895 ± 0.006 | 0.635 ± 0.017 | 0.605 ± 0.011 * | 0.620 ± 0.011 * |
| Train_V1 on Test_V2 | 0.877 ± 0.004 | 0.758 ± 0.012 | 0.811 ± 0.019 | 0.901 ± 0.008 | 0.784 ± 0.008 | 0.941 ± 0.003 | 0.691 ± 0.014 | 0.481 ± 0.012 | 0.586 ± 0.011 |
| Train_V2 on Test_V2 | 0.878 ± 0.006 | 0.759 ± 0.014 | 0.819 ± 0.016 | 0.901 ± 0.008 | 0.787 ± 0.010 | 0.942 ± 0.004 | 0.698 ± 0.024 | 0.485 ± 0.017 | 0.591 ± 0.019 |

*, **, and *** indicate the models trained using V2_Train had significant improvement ($p < 0.05$, $p < 0.01$, and $p < 0.001$, respectively) on V1_Test compared to those trained using V1_Train. †, ††, and ††† indicate the models trained using V2_Train had significant improvement ($p < 0.05$, $p < 0.01$, and $p < 0.001$, respectively) on V2_Test compared to those trained using V1_Train.

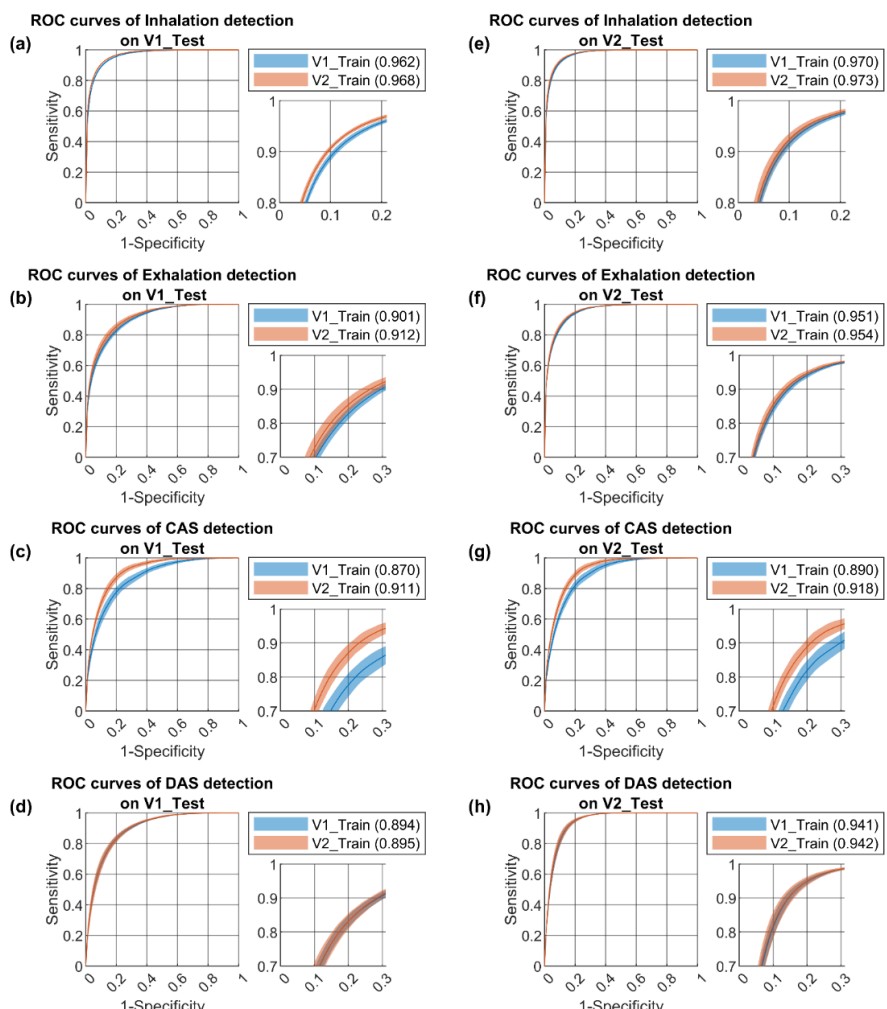

**Figure 6.** ROC curves of the V1_Train-based and V2_Train-based models tested using V1_Test for (**a**) inhalation, (**b**) exhalation, (**c**) CAS, and (**d**) DAS segment detection and using V2_Test for (**e**) inhalation, (**f**) exhalation, (**g**) CAS, and (**h**) DAS segment detection. Solid lines represent the ROC curves, and the colored shades represent areas between the 95% confidence intervals. The blue lines and blue shades stand for the models trained using V1_Train, and the red lines and red shades stand for the models trained using V2_Train. The values in the parentheses in the legends indicate the mean AUC.

### 3.3. Review of Label Quality

Figure 7 illustrates the spectrograms (top graphs in Figure 7a–e) and labels (bottom graphs in Figure 7a–e) of five lung sound recordings. The quality of I and E labels was acceptable. In most cases, inhalations and exhalations can be clearly heard and identified, such as in the example displayed in Figure 7a. However, we can occasionally see wrong labels, such as the exhalation indicated by the white arrow in Figure 7b, or debatable labels, such as the exhalations indicated by the blue arrows in Figure 7b. An exhalation was sometimes not identifiable, such as the sound displayed in Figure 7c.

However, the quality of C labels was unsatisfactory. CASs were often found unlabeled, such as the ones (green arrows) in Figure 7c. In some cases, the labelers were unsure whether to label a borderline case. For example, the labeler did not label the sound indicated by the first green arrow in Figure 7d but labeled the sounds indicated by the second and third green arrows although these sounds did not form clear streak patterns on the spectrogram.

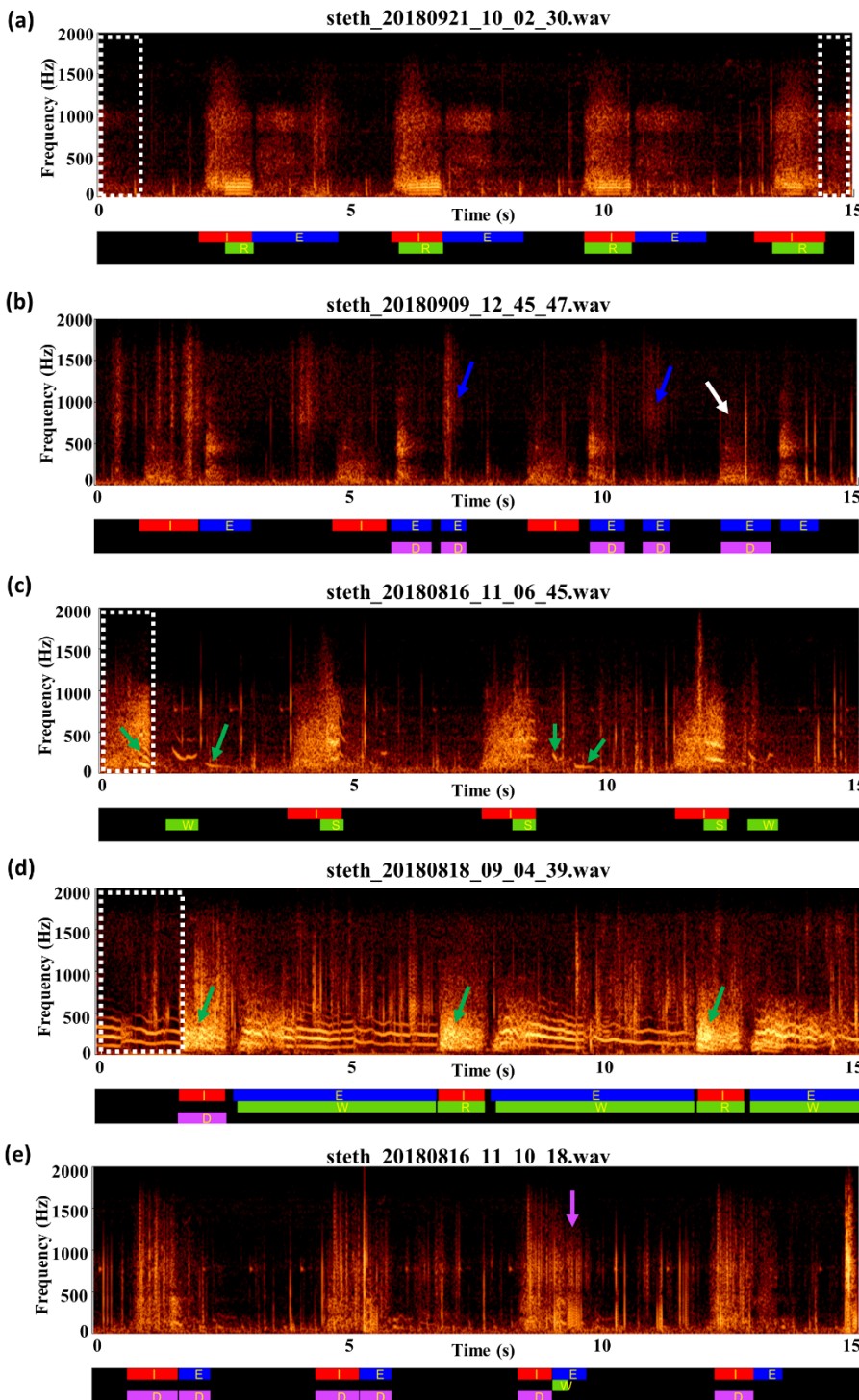

**Figure 7.** Spectrograms (top) and labels (bottom) of five lung sound recordings. In the label graphs at the bottom, red squares with the letter "I" represent I labels. By the same logic, blue squares with the letter "E" represent E labels, purple squares with the letter "D" represent D labels, and green squares with the letters "S," "W," and "R" represent S, W, and R labels, respectively. The blue arrows in (**b**) indicate two sounds with debatable E labels. The white arrow in (**b**) indicates that an inhalation was incorrectly labeled as E. The green arrows in (**c**) indicate CASs that were left without a correct S, W, or R label. The green arrows in (**d**) indicate the sound recordings that did not form clear streak patterns. The purple arrow in (**e**) indicates some noises that interfered with DAS labeling. The white dashed squares in (**a,c,d**) indicate sound events that were left without a label at the beginning and end of a lung sound recording.

Regarding the quality of D labels, even experts could not arrive at a consensus on the correct labeling of a given DAS. For instance, many noises may have DAS-like patterns and may interfere with the labeling process (purple arrow in Figure 7e). Furthermore, the DAS patterns in Figure 7b are sometimes not as clear as those displayed in Figure 7e.

The density plots of the start and end times of I, E, C, and D labels in the V2_Train and V2_Test are displayed in Figure 8. The density of the start and end times of the I (blue lines in Figure 8a–d), E (orange lines in Figure 8a–d), C (green lines in Figure 8a–d), and D (red lines in Figure 8a–d) labels was lower at the beginning and end of the lung sound recordings. However, the density curves reached a higher plateau roughly between 1–14 s for all I, E, C, and D labels in the V2_Train (blue, orange, red, and green lines in Figure 8a,b), though the density curves of I, E, and D labels fluctuated between 1–14 s in the V2_Test (blue, orange, and green lines in Figure 8c,d).

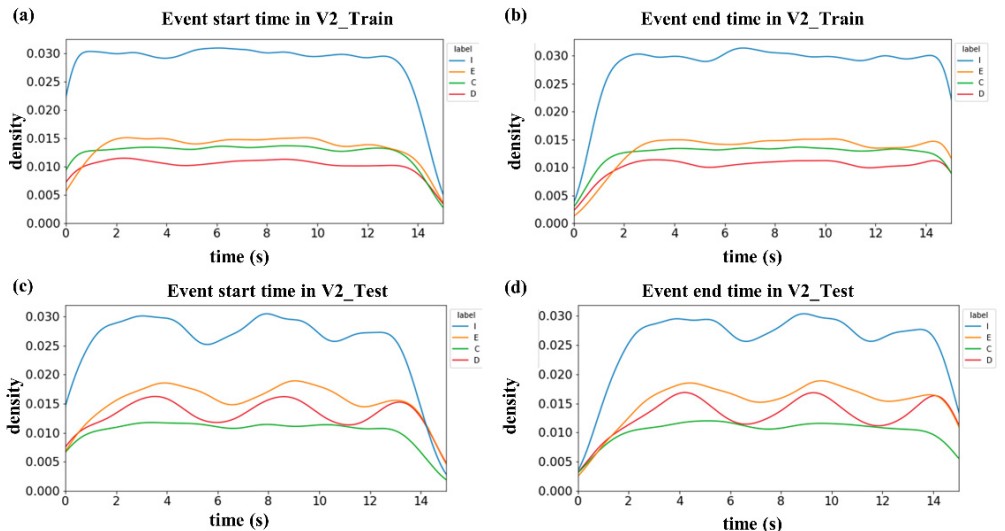

**Figure 8.** Density plots of the start and end times of the I (blue line), E (orange line), C (green line), and D (red line) labels in HF_Lung_V2. (**a**) Start time in V2_Train. (**b**) End time in V2_Train. (**c**) Start time in V2_Test. (**d**) End time in V2_Test.

### 3.4. Overlap Ratios between Labels

Table 5 lists the ratios that indicate the extent to which specific types of labels overlapped with the other types of labels in the V2_Train and V2_Test; 97.7% (58.4% + 39.3%) of the duration of D labels overlapped with the I and E labels in V2_Train, and 96.7% (66.1% + 30.6%) of the duration of D labels overlapped with the I and E labels in V2_Test.

**Table 5.** Overlap ratios between I, E, C, and D labels in HF_Lung_V2.

| Label | Overlapped with | Overlap Ratio | |
|---|---|---|---|
| | | Train | Test |
| I | E | 0.001 | 0.001 |
| | C | 0.179 | 0.167 |
| | D | 0.198 | 0.287 |
| E | I | 0.003 | 0.002 |
| | C | 0.234 | 0.212 |
| | D | 0.283 | 0.266 |
| C | I | 0.463 | 0.517 |
| | E | 0.286 | 0.329 |
| | D | 0.006 | 0.009 |
| D | I | 0.584 | 0.661 |
| | E | 0.393 | 0.306 |
| | C | 0.007 | 0.006 |

## 4. Discussion

The present study reports the process of expanding HF_Lung_V1 to HF_Lung_V2. Performance of the CNN–BiGRU model showed improvement in inhalation, exhalation, CAS, and DAS detection accuracy as the data size increased in HF_Lung_V2, although the improvement in DAS detection may not be significant. The improvement can be clearly observed in Table 4: the models trained on the basis of V2_Train performed better on both V1_Test and V2_Test compared to those trained on the basis of V1_Train. The improvement in segment detection can also be observed in Figure 5: the ROC curves and the 95% confidence intervals of the models trained using V2_Train (red lines and red shades) located closer to the left upper corner than those trained using V1_Train (blue lines and blue shades).

Small data regions, small power-law regions, and irreducible error regions are present in the curve of the power law of learning [30]. The generalization error (log-scale) decreases as the training set size (log-scale) in the power-law region increases [30]. We did not investigate whether the increase in the size of HF_Lung_V2 (which is 1.43 times that of HF_Lung_V1) was in the power-law region. The promising improvement in the performance of inhalation, exhalation, CAS, and DAS detection encourages us to continue collecting more breathing lung sounds and to build a larger dataset.

Notably, the benchmark performance of the CNN-BiGRU models trained using V1_Train and tested on V1_Test differed from those reported in our previous study [18], which is due to retraining the models. Retraining was performed because event detection evaluation was performed somewhat differently. Specifically, unlike in our previous work [18], ground-truth event labels and the results of event prediction were used alternately as references. Thus, an FP event and an FN event could have been counted heedlessly within the same period. Furthermore, five-fold cross-validation was repeated thrice, and the measurements were averaged and reported in this study.

Label quality review revealed that the quality of I and E labels was acceptable. However, the quality of C labels was unsatisfactory. This may have occurred because we asked the labelers to perform the labeling mostly on the basis of what they heard and to use the spectrogram only as an aid. We also asked them not to label a sound without being sure the label was appropriate. Furthermore, some noisy labels may have resulted from mistakes made in operating the labeling software, such as the E label indicated by the white arrow in Figure 7b. In addition, it is especially difficult to establish clear criteria to decide whether a sound can be counted as a CAS or to which types of CAS a sound belongs. Many borderline rhonchi did not form a clear streak pattern; thus, the labelers were unsure whether to label them (green arrows in Figure 7d). Upon the exclusion of R labels from the C labels, the model exhibited superior performance in CAS detection (data not shown). As for D labels, the evaluation quality was strongly influenced by the indifferentiable rubbing-sound noises. The labelers also tended to ignore the sounds located at the very beginning and end of an audio file because the event was cut off or because the full duration of an event was not heard or observed (white squares in Figure 7a,c,d). This phenomenon can be clearly observed in the density plots (Figure 8a–d), which show that the density of the start and end times of labels was lower at the beginning and end of the audio files. This is another source of noisy labels that may cause confusion during CNN training. Additionally, the lower density at the beginning and end of an audio file may cause some false negative events in real use of the models, which is associated with the use of BiGRU networks in our models. The fluctuation of the density curves of the start and end times of the I, E, and D labels between the 1–14 s in the V2_Test (Figure 7c,d) probably resulted from smaller numbers of labels compared to those in the V2_Train.

DAS detection performance was greatly influenced by sound overlapping. We labeled only the start and end times of a series of DASs (mean duration of each series: 0.86 s; Table 2); however, each DAS, i.e., the mean duration of fine and coarse crackle sounds: approximately 5 and 15 ms [6], respectively, was not labeled. This caused the overlapping of more than 96% of D labels with the I and E labels (Table 5). Moreover, because only audio

recordings containing D labels were used to train the DAS detection model, the trained model showed a tendency to incorrectly identify inhalations and exhalations as DASs. A new dataset that contains more balanced data or a new training strategy is required to solve this problem in future studies.

**Supplementary Materials:** The following supporting information can be downloaded at: https://www.mdpi.com/article/10.3390/app12157623/s1, Table S1: Characteristics of co-occurring diseases in the subjects in HF_Lung_V2; Table S2: Statistics of lung sound files and labels in the HF_Lung_V1 and HF_Lung_V2 databases grouped by recording devices.

**Author Contributions:** Conceptualization, F.-S.H., S.-R.H. and Y.-R.C.; methodology, S.-R.H., C.-W.H., Y.-R.C. and C.-C.C.; formal analysis, C.-W.H. and C.-C.C.; resources, J.H. and C.-W.C.; writing—original draft preparation, F.-S.H. and S.-R.H.; writing—review and editing, F.-S.H., S.-R.H. and F.L.; project administration, F.-S.H. and F.L.; funding acquisition, F.-S.H.; investigation, J.H. and C.-W.C. All authors have read and agreed to the published version of the manuscript.

**Funding:** This study is supported by the Ministry of Science and Technology, Taiwan, R.O.C. under Grant no. MOST 109-EC-17-A-22-I3-0009.

**Institutional Review Board Statement:** The study was conducted in accordance with the Declaration of Helsinki and the lung sound recording was approved by the Institutional Review Board of Far Eastern Memorial Hospital on 19 August 2018 (protocol code 107052-F).

**Informed Consent Statement:** Written informed consent was obtained from all subjects involved in the study (patients or patients' legal representatives). Written informed consent has been obtained from the patients to publish this paper.

**Data Availability Statement:** HF_Lung_V1, including lung sound recordings and labels, can be downloaded at https://gitlab.com/techsupportHF/HF_Lung_V1 (accessed on 18 January 2022). The lung sound recordings in HF_Lung_V1_IP can be downloaded at https://gitlab.com/techsupportHF/HF_Lung_V1_IP (accessed on 13 April 2022). However, the labels in HF_Lung_V1_IP are proprietary property owned by Heroic Faith. Please send an email to fshsu@heroic-faith.com to request access to the labels.

**Acknowledgments:** Taiwan's Raising Children Medical Foundation sponsored the lung sound collection activity. Heroic Faith Medical Science Co., Ltd. sponsored the data labeling and deep learning model training. The authors thank Taiwan's National Center for High-Performance Computing for providing the computing resources. The authors also thank the employees of Heroic Faith Medical Science Co., Ltd., who contributed to the establishment of the HF_Lung_V2 database.

**Conflicts of Interest:** F.-S.H., S.-R.H. and Y.-R.C. are employees of Heroic Faith Medical Science Co., Ltd. C.-W.H. and C.-C.C. are with Avalanche Computing Inc., who are commissioned by Heroic Faith Medical Science to train the AI models.

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
