# Peer review of "A Progressively Expanded Database for Automated Lung Sound Analysis: An Update"

_applsci, doi:10.3390/app12157623_

Round 1
Reviewer 1 Report
I thank the Editor for the opportunity to review the article from Taiwanese colleagues. In fact, it is a beautiful article both in terms of scientific content and stylistic appeal. Overall, it is a well structured article which is basically ready for publication; however, I would like to share some little comments about it:
- the text is clear in all sections and above all in the methodology. Biases are really minimized and Authors prepared and reported a well-structured analysis on a scientific level.
- the benchmarking session 2.5 appears extremely specialized and well beyond my possibilities, therefore I do not express myself on this session, leaving it to other reviewers.
- table 2 is adequate; for a better glance it might be useful to perform a ROC curve analysis, possibly a cumulative ROC curves with all values of V2, which allows to quickly evaluate the average, minimum and maximum accuracy of V2 dataset. The same applies to perform a superimposed ROC curve between V1 and V2, with the significance inserted between the two curves.
Reviewer 2 Report
The article presents several strengths, starting from the purpose and the topicality of the topic, among other things the authors have collected a very good bibliography, it is full of illustrative tables and images that help the reader to decipher the data presented; finally, the use of the English language is really excellent and convincing!
A very beautiful, exciting work, never read before; it has a good case history
These are the observations I feel I can give
Reviewer 3 Report
General comment:
This work deals with the exentesion of an existing open-access dataset of lung sounds.The new data are also used to study noisy labels and overlapping and their impact on the training process.
The work appears to be incremental. Novelty and differences wrt previous work must be stressed and highlighted.
Specific comments throughout the paper:
1. Introduction
Lines 50-54: Please revise this part beacuse the writing is not so clear.
Line 55:"to using" - Too many errors related to the English language are present. The manuscript needs a thorough proofreading.
Line 57: I am missing the introduction of references from [14] to [18]. There is a gap. Please revise the reference numbering.
Lines 65-86: The differences between the authors' database and the others taken from the literature are not described and clear. Please provide a coherent discussion.
2. Materials and Methods
Lines 96-99: Missing details about the statistical analysis of the database. What about the age, sex, existing pathologies, etc? These information are not reported. Please provide them and report mean ± standard dev of any useful data.
Line 136: "board-certified" - by who and how? Please provide additional details.
Lines 147-151: Examples of the signals and labels must be provided in a figure. This would add value and quality to the work.
Lines 152-157: Good for the training strategy. However, the number of subjects, number of audio tracks and trainint/tests numerosity are missing details. Only the ratio is given. Please provide all the methodological information which are required to ensure the reproducibility of your approach.
3. Results
Please check the style of the section and subsection titles.
Fig. 5: The size of the font of the x- and y-axes can be increased to get a more readable image.
4. Discussion
The discussion section is appreciated.
Round 2
Reviewer 3 Report
I sincerely thank the authors for providing exhaustive replies to my questions and doubts.
They thoroughly revised their work and significantly improved it in terms of referencing, methodological aspects and results presentation.